# Cutin from *Solanum Myriacanthum* Dunal and *Solanum Aculeatissimum* Jacq. as a Potential Raw Material for Biopolymers

**DOI:** 10.3390/polym12091945

**Published:** 2020-08-28

**Authors:** Mayra Beatriz Gómez-Patiño, Rosa Estrada-Reyes, María Elena Vargas-Diaz, Daniel Arrieta-Baez

**Affiliations:** 1Instituto Politécnico Nacional-CNMN, Unidad Profesional Adolfo López Mateos, Col. Zacatenco, México City CDMX CP 07738, Mexico; bethzem86@gmail.com; 2Laboratorio de Fitofarmacología, Dirección de Investigaciones en Neurociencias, Instituto Nacional de Psiquiatría Ramón de la Fuente Muñiz, Calzada México-Xochimilco 101, San Lorenzo Huipulco, Tlalpan, Ciudad de México 14370, Mexico; restrada@imp.edu.mx; 3Instituto Politécnico Nacional-Departamento de Química Orgánica, Escuela Nacional de Ciencias Biológicas, Prolongación de Carpio y Plan de Ayala S/N, Colonia Santo Tomás D.F. 11340, Mexico; evargasvd@yahoo.com.mx

**Keywords:** cutin, cuticles, bioplastics, biopolymers, solanum: CPMAS ^13^C NMR

## Abstract

Plant cuticles have attracted attention because they can be used to produce hydrophobic films as models for novel biopolymers. Usually, cuticles are obtained from agroresidual waste. To find new renewable natural sources to design green and commercially available bioplastics, fruits of *S. aculeatissimum* and *S. myriacanthum* were analyzed. These fruits are not used for human or animal consumption, mainly because the fruit is composed of seeds. Fruit peels were object of enzymatic and chemical methods to get thick cutins in good yields (approximately 77% from dry weight), and they were studied by solid-state resonance techniques (CPMAS ^13^C NMR), attenuated total reflection-Fourier transform infrared spectroscopy (ATR-FTIR), atomic force microscopy (AFM) and direct injection electrospray ionization mass spectrometry (DIESI-MS) analytical methods. The main component of *S. aculeatissimum* cutin is 10,16-dihydroxypalmitic acid (10,16-DHPA, 69.84%), while *S. myriacanthum* cutin besides of 10,16-DHPA (44.02%); another two C18 monomers: 9,10,18-trihydroxy-octadecanoic acid (24.03%) and 18-hydroxy-9S,10R-epoxy-octadecanoic acid (9.36%) are present. The hydrolyzed cutins were used to produce films demonstrating that both cutins could be a potential raw material for different biopolymers.

## 1. Introduction

The cuticle is the outer membrane that covers the aerial parts of plants, such as the stem, leaves, flowers, and fruit. In the evolution of plants, the cuticle plays a critical role against the loss of water from internal tissues [1,2,3]. In the same way, this biopolymer plays an essential physiological role as it is considered a first barrier that prevents the entry of pathogens and pesticides [4,5]. The cuticle consists primarily of cutin (a C16 and C18 long-chain hydroxy acid polyester), cell wall polysaccharides (cellulose, hemicellulose, and pectin), as well as epicuticular fatty acids [6,7,8]. Cutin, and other important natural biopolymers such as lignin, cellulose, and chitin, has been shown to have important bioplastic properties [9,10,11,12,13]. For this reason, they have been considered as models for plastic materials with biodegradable characteristics that eventually could replace conventional plastics derived from petroleum in specific industrial uses [14].

Raw renewable materials have a high impact on the cost of bio-based plastic production and, in this regard, different efforts have been directed to use biomass to get or produce biopolymers. Biopolymers such as starch, cellulose, lignocellulosic materials, and proteins; bio-derived monomers like polylactic acid (PLA), polyglycolic acid (PGA), and biodegradable polymers from petrochemicals (aliphatic polyesters, aromatic co-polyesters and polyvinyl alcohols) have been investigated as sources for bioplastics [15,16].

In this sense, the cutins of some fruits for human consumption, such as tomatoes, citrus have shown good physicochemical characteristics as biopolymers [9,17,18,19,20]. However, the ethical problem that could be generated has led researchers to use industrial waste products. In fact, the vegetable food processing industry generates a significant amount of waste worldwide [21]. Thus, from agro-industrial residues, biomaterials have been generated, and some of them have been used in the food packaging industry, and other bioplastic applications [22,23].

In the present work, we have searched for fruits that are not for human consumption, which present the same chemical characteristics of the cuticles used for biopolymers applications in order to be considered promising candidates as a raw material to produce bioplastics. In this sense, the cuticles of the fruits of two species of the *Solaneum* genus were studied. *S. aculeatissimum* and *S. myriacanthum* are shrubs that grow in the wild, covered with thin spines up to 18 mm long, that produce small fruits of approximately 2–3 cm, which are mainly filled with seeds. *S. aculeatissimum* is native to Brazil, but it could be found in tropical Africa and Asia. In Mexico, it is distributed in the states of Jalisco, Oaxaca, Chiapas, Veracruz, and Puebla [24]. It is considered a toxic plant due to the alkaloids present in the seeds and leaves [25,26]. Its fruit is green when it is immature and red when it is ripe. *S. myriacanthum* is native to central and south America, although it is also distributed in Asia, mainly in India. In Mexico, it is distributed in the states of Chiapas, Veracruz, Oaxaca, and Puebla [24]. Anthelmintic properties are attributed to the extracts of the fruit [27]. Its fruit is green when immature and yellow when ripe.

*S. aculeatissimum* and *S. myriacanthum* cutins were extracted and analyzed by means of CPMAS ^13^C NMR, ATR-FTIR, AFM and DIESI-MS, and their components and physicochemical characteristics were determined and compared with other cutin components. From their hydrolyzed components, films were obtained and characterized and, from these results, both Solanum species could be considered as a raw material for biopolymers used in different fields of the plastic industry.

## 2. Materials and Methods

### 2.1. Chemicals

Trifluoroacetic acid (TFA), KOH, and other reagents were purchased from Sigma-Aldrich (St. Louis, MO, USA). The enzymes *Aspergillus niger* pectinase (EC 3.2.1.15) (specific activity ≥ 5 unit/mg protein), *A. niger* cellulase (EC 3.2.1.4) (specific activity ≥ 0.3 units/mg protein) and *A. niger* hemicellulose (EC 3.2.1.4) (specific activity 2.3 units/mg protein) were purchased from Sigma Chemicals (St Louis, MO, USA).

### 2.2. Isolation of Cutin

*S. aculeatissimum* and *S. myriacanthum* fruits were collected in Cuetzalan, Puebla (20°01′48.7″ N 97°29′04.3″ W) in September 2018. Fruits were washed with tap water, cut, and the seeds were removed to obtain the cuticle. The cutin was obtained using a previously reported protocol [9]. Briefly, the cuticle was treated with *A. niger* pectinase (EC 3.2.1.15, St Louis, MO, USA) (10 mg/mL) for 1 week. After this, cell wall polysaccharides were removed with an enzymatic digestion using *A. niger* cellulose (EC 3.2.1.4, St Louis, MO, USA) (80 mg/mL) for 1 week and *A. niger* hemicellulose (EC 3.2.1.4, St Louis, MO, USA) (80 mg/mL) for 1 week. To complete the extraction, a Soxhlet procedure was done with methylene chloride:methanol (1:1 *v*/*v*, 48 h, St Louis, MO, USA) to remove residual compounds of cutin such as monosaccharides and waxes. Five hundred grams of *S. aculeatissimum* dried peel yielded 386 g (77.2%) of cutin and 500 g of *S. myriacanthum* dried peel yielded 394 g (78.8%) of cutin. The resulting cutins were analyzed by Cross Polarization Magic-Angle Spinning (CPMAS ^13^C NMR), attenuated total reflection-Fourier transform infrared spectroscopy (ATR-FTIR) and atomic force microscopy (AFM).

### 2.3. Treatment of Cutin with Trifluoroacetic Acid (TFA)

One hundred and fifty milligrams of the obtained cutin from the *S. aculeatissimum* or *S. myriacanthum* fruits was added to an aqueous TFA solution 2.0 mol·L^−1^ and stirred at 115 ± 5 °C for 2 h in separated experiments. Each reaction was filtered, and the insoluble material was washed using chloroform-methanol (1:1, *v*/*v*, St Louis, MO, USA) for 1 h to obtain the TFA-hydrolyzed cutin (TFA-HC). The TFA-HC was separated by filtration, dried, and analyzed by CPMAS ^13^C NMR. The TFA solution was co-evaporated with methanol and the resulting solids were redissolved in methanol to give a clear brown solution, which was later analyzed by DIESI-MS (Bruker Daltonics, Biellerica, MA, USA) and solution-state NMR (Billerica, MA, USA) [28].

### 2.4. Alkaline Hydrolysis of the Cutin with KOH/MeOH

Fifty milligrams of *S. aculeatissimum* or *S. myriacanthum* cutin was added to 50 mL of 1.5 mol·L^−1^ methanolic KOH solution, and the mixture was stirred at room temperature for 24 h. After this time, the reaction was filtered, neutralized, and monomers were extracted with CHCl_3_-MeOH. The dried extract was weighed, dissolved in CHCl_3_-MeOH and analyzed by DIESI-MS (Bruker Daltonics, Biellerica, MA, USA).

### 2.5. Preparation of Cutin Films

Twenty-five milligrams of hydrolyzed *S. aculeatissimum* or *S. myriacanthum* cutin were added to 5 mL of ultrapure methanol:chloroform (1:1, *v*/*v*, St Louis, MO, USA) solution. The solution was sonicated for 30 s and it was deposited in plastic Petri dishes for making films using the casting method. On the other hand, 5 µL of the solution were deposited on a watch glass to study the structures of self-assembled layers. After this, films were kept in a chemical hood to remove residual solvents from the films [29].

### 2.6. NMR Spectroscopy

*S. aculeatissimum* and *S. myriacanthum* cutin and TFA-HC were analyzed using standard CPMAS ^13^C NMR experiments carried out on a Varian Instruments Unityplus 300 widebore spectrometer (Palo Alto, CA, USA) equipped for solid-state NMR. The resonance frequency was 74.443 MHz, with a customary acquisition time of 30 ms, a delay time of 2 s between successive acquisitions and a CP contact time of 1.5 ms. Typically, each 30 mg sample was packed into a 5 mm rotor and supersonic MAS probe from Doty Scientific (Columbia, SC, USA), then spun at 6.00 (±0.1 kHz) at room temperature for approximately 10 h. No spinning sidebands were observed upon downfield from the major carbonyl, aromatic, or aliphatic carbon peaks, presumably due to motional averaging and/or excessive broadening of such features.

Soluble products derived from the TFA hydrolysis were examined using ^1^H NMR. Experiments were conducted on a Bruker Instruments ASCEND 750 spectrometer (Billerica, MA, USA). The resonance frequency was 750.12 MHz, with a typical acquisition time of 2.1845 s and a delay time of 1.0 s between successive acquisitions. The ^1^H and ^13^C chemical shifts are given in units of δ (ppm), using tetramethylsilane (TMS) as internal standard.

### 2.7. ATR-FTIR Spectroscopy

Attenuated Total Reflectance Fourier transform infrared spectroscopy (ATR-FTIR) spectra were recorded with a BOMEM 157 FTIR spectrometer (Bomem Inc., Quebec, Canada) equipped with a deuterated triglycinesulfate (DTGS) detector. The instrument was under a continuous dry air purge to eliminate atmospheric water vapor. The spectra were recorded in the region of 4000 to 400 cm^−1^.

### 2.8. Atomic Force Microscopy

The samples for the Atomic Force Microscopy analysis (AFM) were prepared by fixing to a metallic disk with double-sided tape. The images themselves were taken using a MultiMode AFMV (Bruker, SantaBarbara, CA, USA) in air with an RTESP cantilever, and operating the AFM in tapping mode. The size of each image was 5 × 5 µm^2^. The roughness parameters *R*_q_ and *R*_a_ were determined using the expressions *R*_q_ = √ΣZ21 = N and *R*_a_ = 1 = NΣNj = 1 jZjj, where *R*_q_ is the root mean square average of the height deviations, Ra is the arithmetic average of the absolute values of the surface height deviations, Z is the height value, and N is the number of data points. These parameters were obtain using the NanoScope Analysis image software.

### 2.9. Mass Spectrometry

Direct Ionization analysis (DIESI-MS) was done on a Bruker MicrOTOF-QII system, using an electrospray ionization (ESI) interface (Bruker Daltonics, Biellerica, MA, USA) operated in the negative ion mode. A solution of 10 µL of the sample resuspended in 1 mL of methanol was filtered with a 0.25 µm polytetrafluoroethylene (PTFE) filter and diluted 1:100 with methanol. Diluted samples were directly infused into the ESI source and analyzed in negative mode. Nitrogen was used with a flow rate of 4 L/min (0.4 Bar) as a drying and nebulizer gas, with a gas temperature of 180 °C and a capillary voltage set to 4500 V. The spectrometer was calibrated with an ESI-TOF tuning mix calibrant (Sigma-Aldrich, Toluca, Estado de México, México).

MS/MS analysis was performed using negative electrospray ionization (ESI^−^), and the obtained fragments were analyzed by a Bruker Compass Data Analysis 4.0 (Bruker Daltonics, Technical Note 008, 2004, Bruker Daltonics, Biellerica, MA, USA). An accuracy threshold of 5 ppm was established to confirm the elemental compositions.

## 3. Results

*S. aculeatissimum* and *S. myriacanthum* fruits were collected in the Cuetzalan, Puebla (México) region (Figure 1). Usually, the mature fruit is a globose berry 2–3 cm in diameter, and it is composed of the peel (≈0.2 mm thick) that represent a 25–30% percent of the fruit, 10% of a polysaccharides layer, and 50–60% of seeds. Once the peels were washed and dried, cutins were obtained by previously published methods. Cutin obtained from the dry peels were in very high yield (≈77%) in relation to other fruit cutins, such as tomato or citrus fruits (≈0.5%) [9,30]. Most of the compounds hydrolyzed with the enzymatic treatment were identified as monosaccharides (Glu and Fru, data not shown), and cutins were characterized by solid-state resonance techniques (CPMAS ^13^C NMR), ATR-FTIR, and AFM.

### 3.1. CPMAS ^13^C NMR Analysis

Cutins obtained from *S. aculeatissimum* and *S. myriacanthum* were analyzed by CPMAS ^13^C NMR, and their spectra are shown in Figure 2. According to our previous studies in fruit cuticles, typical resonances of aliphatic-aromatic polyesters are exhibited: bulk methylenes (20–35 ppm), oxygenated aliphatic carbons (55–85 ppm), aromatics and olefins (105–155 ppm), and carbonyl groups (172 ppm) signals.

Some of these signals belong to the carbohydrate moieties (C6 at 60 ppm, C2,3,5 at 70–75 ppm, C4 at 83 ppm, and C1 at 101–105 ppm), some of these peaks could overlap with oxygenated aliphatic signals. However, every cutin showed unique NMR characteristics: more aromatic peaks are evident in *S. aculeatissimum* (Figure 2, upper spectrum), while in *S. myriacanthum* (Figure 2, lower spectrum) peaks at 52 and 56 ppm are present.

To garner more information about these materials, cutins from both fruits were the object of a TFA hydrolysis. It has been demonstrated that TFA hydrolysis could be used to remove non-cellulosic polysaccharides with the advantage that TFA is easy to remove by evaporation rather than a loss-prone neutralization step [28].

*S. aculeatissimum* cutin was found to be resistant to TFA hydrolysis. There was a minimal weight loss, and as seen in Figure 3, and most of the peaks remain as in the spectra without TFA treatment. However, *S. myriacanthum* shows ≈ 8% of weight loss, and this can be attributed to the disappearance of compounds with peaks at 50 ppm. The soluble part obtained from the TFA hydrolysis was studied, and it was found that these peaks belong to an epoxidated C18 long-chain aliphatic acid (see Appendix A).

### 3.2. Infrared Spectroscopy Analysis of the S. Aculeatissimum and S. Myriacanthum Cutins

Isolated cutins have been characterized in situ at their functional chemical groups as well as their interactions at the cuticular levels with exogenous chemicals [8,31]. The ATR FT-IR analysis of *S. aculeatissimum* and *S. myriacanthum* cutins (Figure 4) were characterized as follows: hydroxyl groups of the polysaccharide domain and residual carboxylic acids showed its absorption maxima as broadband at 3860 cm^−1^, characteristic intense bands corresponding to the asymmetrical and symmetrical stretching vibrations of the methylene CH_2_ region at 2905 and 2850 cm^−1^, with the bending vibrations at 1462 and 1350 cm^−1^, which came from the aliphatic components present in the cutin. Another group of signals associated with the cutin matrix is that from 1600 to 1750 cm^−1^ attributed to the carbonyl C=O stretching band in ester groups, and their asymmetric stretching vibrations of C–CO–O at 1100 cm^−1^. These assignations agree with those reported and used in the study of non-isolated plant cutins [8].

Figure 4 shows that IR spectra for both cutins are very similar, and the most intense bands correspond to the main domains of this polyester: aliphatic and polysaccharides groups. However, two groups of signals are making the difference between them. For *S. aculeatissimum* cutin, a group of bands at 1500 to 1650 cm^−1^ related to aromatic and C=C functional groups are less intense in *S. myriacanthum* cutin, due to the low presence of aromatics. On the other hand, the group of signals at 1100 cm^−1^ is broader and more intense in *S. myriacanthum*, possibly because of a poliesterification with at least two different long-chain acids. These observations agreed with the CPMAS ^13^C NMR analysis.

### 3.3. Atomic Force Microscopy Analysis

The cuticles obtained from the fruits of *S. aculeatissimum* and *S. myriacanthum* were analyzed through AFM. Figure 5 shows that these cutins are thicker than other fruit cutins, such as tomatoes, lemon, orange. The AFM amplitude error images showed that cutin surfaces are composed mainly of fibers that give the characteristic roughness (Figure 5C,D). The fibers are more homogeneous in the *S. aculeatissimum* cutin with an average thickness of 34 nm, while in the *S. myriacanthum* cutin they are irregularly present, with fibers ranged from 125 to 23 nm that were observed. The roughness study showed that *S. aculeatissimum* has a *R*_q_ of 1.8 nm and a *R*_a_ of 1.3 nm, while the cutin of *S. myriacanthum* showed a lower roughness with a *R*_q_ of 3.4 nm and a *R*_a_ of 2.6 nm.

### 3.4. Alkaline Hydrolysis (KOH/MeOH)

To study the main aliphatic components present in the *S. aculeatissimum* and *S. myriacanthum* cutins, a complementary analysis was done with alkaline hydrolysis. In both cases, around ≈93% of the cuticular material was hydrolyzed. Soluble products from the alkaline hydrolysis were analyzed by means of direct-injection electrospray ionization mass spectrometry in negative mode (DIESI-MS, see Appendix A) and the compounds identified by the *ms/ms* analysis, are reported in Table 1.

Even when most of the compounds are present in both cutins, some differences can be observed. The main constituent identified in *S. aculeatissimum* cutin was 10,16-dihydroxyhexadecanoic acid (10,16-DHPA), an important monomer present in different cutins such as tomato, citrus cuticles and green pepper [32], in a 69.84% of the relative abundance. Aromatic and some derivatives compounds were detected in agreement with the CPMAS ^13^C NMR data. 10,16-DHPA was found in *S. myriacanthum* cutin. However, two other significant monomers are present: 9,10,18-trihydroxy-octadecanoic acid and 18-hydroxy-9S,10R-epoxy-octadecanoic acid in 24.03 and 9.36%, respectively. According to the TFA-hydrolysis analysis, the epoxilated long-chain aliphatic acid was hydrolyzed and obtained almost pure, according to the NMR analysis (see Appendix A). This observation could suggest that it is present in a different domain from the other components. The predominance of C16 long-chain acids in cutins is very common and corroborates previous cutin reports. However, it is important to highlight that most of the 25% of the main monomers in *S. myriacanthum* cutin are C18 acids. The presence of these C16 and C18 monomers could be the reason for the broadband esterification detected at 100 cm^−1^ in the ATR-FTIR spectrum. Aromatic compounds are not present as in *S. aculeatissimum* cutin, which agrees with the NMR analysis.

### 3.5. Analysis of the Films Prepared from Hydrolyzed Cutins

To demonstrate that *S. aculeatissimum* and *S. myriacanthum* cutins could be a good material for biopolymer, films were prepared by simple blending in solvents. Representative photographs of the films prepared from the hydrolyzed cutins are shown in Figure 6. Both samples have a waxy consistency, but their surface was quite homogeneous. Films were characterized through ATR-FTIR and AFM.

### 3.6. ATR-FTIR Analysis of the Films

Analysis using Fourier transform infrared (FTIR) spectroscopy indicated that films from hydrolyzed cutins keep the spectral features of the original cutins. However, most of the signals demonstrate that bands associated with ester groups disappeared, especially the absorption at 1630 cm^−1^ that belongs to the stretching of C=O of ester groups. However, the presence of the band at 1127 cm^−1^, ascribed to the asymmetric stretching vibrations of C–CO–O, demonstrates that part of this polyester network remains, or monomers were partially polymerized (Figure 7).

### 3.7. AFM Analysis of the Films

Atomic Force Microscopy (AFM) analysis shows a different topography from that observed in the original cutins. There is no occurrence of fibers that could be attributed to the cellulose or pectin presence [33]. According to DIESI-MS analysis, there is not a presence of sugars or some oligo- or polysaccharides in the soluble hydrolyzed cutins. The roughness study showed that film from *S. aculeatissimum* cutin has a value of a *R*_q_ 0.527 nm and a *R*_a_ 0.406 nm, while that obtained from *S. myriacanthum* cutin showed values of *R*_q_ 0.973 nm and Ra 0.584 nm (Figure 8).

The homogeneity could be attributed to a good organization and a high degree of order of the monomers, oligomers or polymers present in the hydrolyzed cutin. This characteristic is highly important to get films with small porous or cavities distributed along the surface.

## 4. Conclusions

In this work, we have demonstrated that *S. aculeatissimum* and *S. myriacanthum* cuticles have a good percentage of cutin—around 70%, from the dry weight, more than other studied fruits such as tomato or citrus fruits (≈0.5%). These cutins have the same monomers composition reported in other fruits cuticles such as tomato, citric fruits, or pepper, were the main component was 10,16-DHPA. Films obtained from the hydrolyzed cutins showed a good homogeneity. Furthermore, the fact that these fruits are not for human or animal consumption makes it feasible for them to be considered as a potential raw material to produce sustainable composite materials as an alternative to traditional plastics.

## Figures and Tables

**Figure 1 polymers-12-01945-f001:**
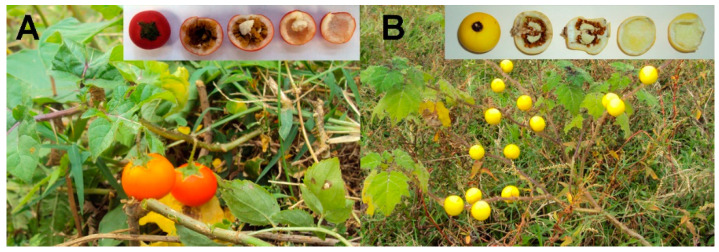
Photographs of the plants and fruits of (**A**) *S. aculeatissimum* Jacq, and (**B**) *S. myriacanthum* Dunal.

**Figure 2 polymers-12-01945-f002:**
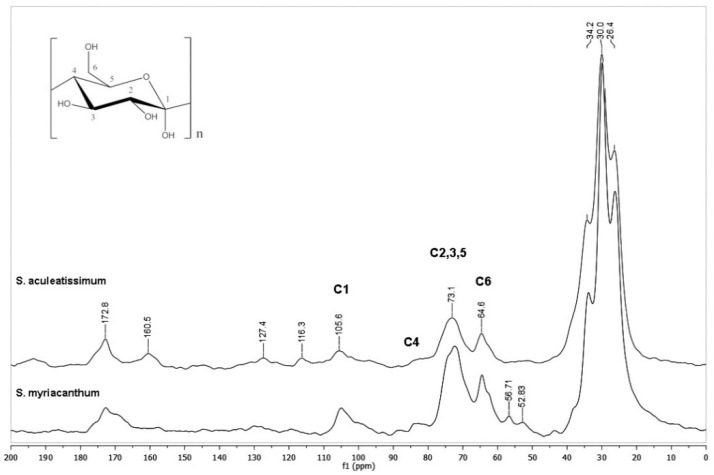
CPMAS ^13^C NMR spectra of cutins from *S. aculeatissimum* and *S. myriacanthum*.

**Figure 3 polymers-12-01945-f003:**
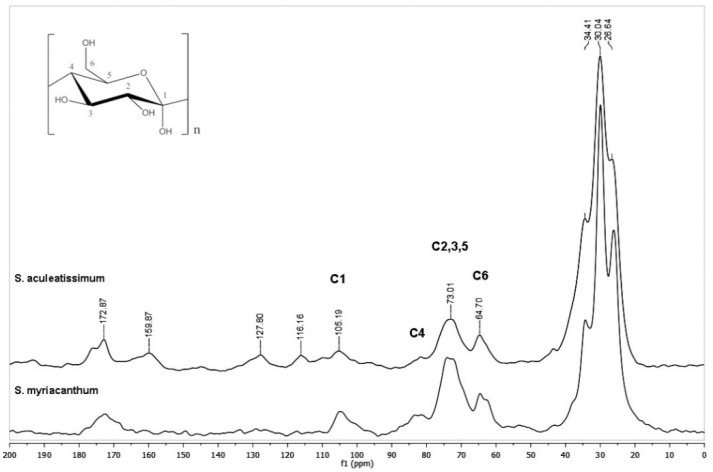
CPMAS ^13^C NMR spectra of cutins from *S. aculeatissimum* and *S. myriacanthum* after trifluoracetic acid (TFA) hydrolysis (TFA-HC).

**Figure 4 polymers-12-01945-f004:**
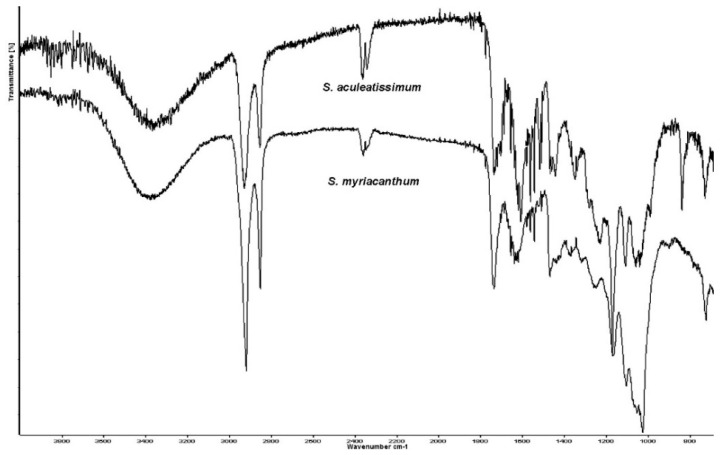
Attenuated Total Reflectance Fourier transform infrared spectroscopy (ATR-FTIR) spectra of *S. aculeatissimum* and *S. myriacanthum*.

**Figure 5 polymers-12-01945-f005:**
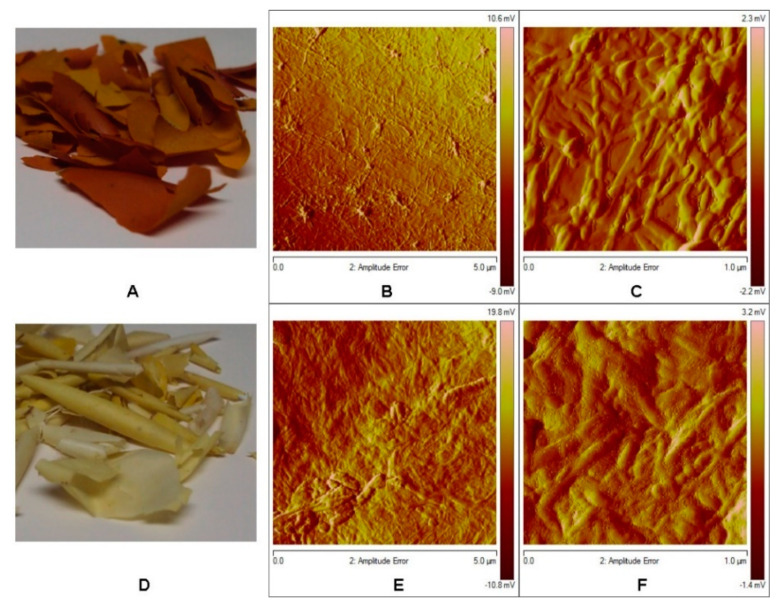
Atomic force microscopy tapping mode topographical images from (**A**) *S. aculeatissimum* (**B**,**C**): 5.0 and 1.0 µm, respectively) and (**D**) *S. myriacanthum* (**E**,**F**): 5.0 and 1.0 µm, respectively).

**Figure 6 polymers-12-01945-f006:**
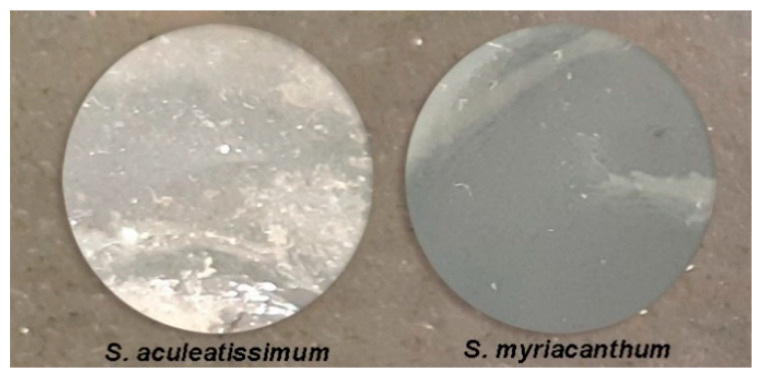
Photographs of the films prepared from hydrolyzed cutins.

**Figure 7 polymers-12-01945-f007:**
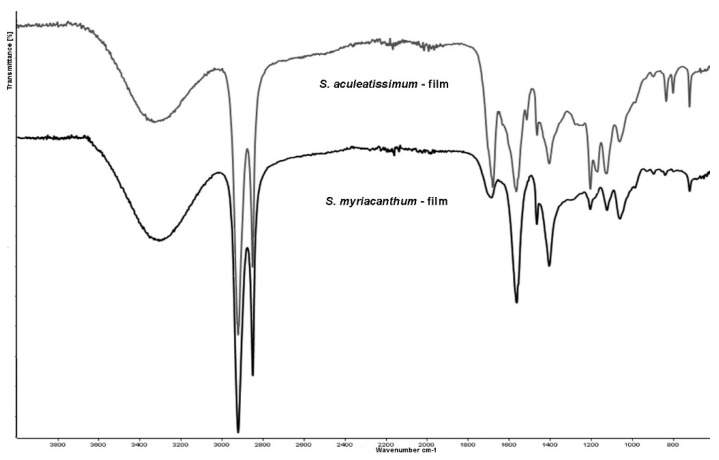
ATR-FTIR spectra of the films from hydrolyzed cutins of *S. aculeatissimum* and *S. myriacanthum.*

**Figure 8 polymers-12-01945-f008:**
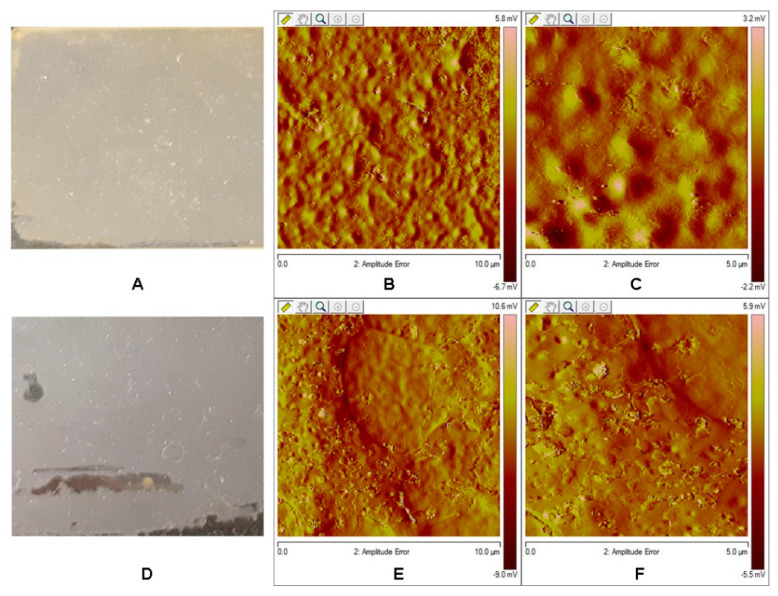
Atomic force microscopy tapping mode topographical images from (**A**) *S. aculeatissimum* (**B**,**C**): 5.0 and 1.0 µm, respectively) and (**D**) *S. myriacanthum* (**E**,**F**): 5.0 and 1.0 µm, respectively).

**Table 1 polymers-12-01945-t001:** Identification of the Main Compounds in the Soluble Fraction of the Alkaline Hydrolysis.

Name	[M − H]^−^_obs_	[M − H]^−^_exact_	Formula	Error	%RA	
					SA	SM
Coumaric acid	163.0386	163.0389	C_9_H_8_O_3_	2.1	0.18	---
Capric acid	171.1352	171.1379	C_10_H_20_O_2_	3.5	0.36	0.25
Coniferaldehyde	177.0557	177.0546	C_10_H_10_O_3_	3.6	0.54	---
n-Nonanedioic acid	187.1001	187.0964	C_9_H_16_O_4_	4.1	0.18	0.25
Ferulic acid	193.0473	193.0495	C_10_H_10_O_4_	2.5	0.18	---
Lauric acid	199.1689	199.1692	C_12_H_24_O_2_	2.5	0.18	0.25
Myristic acid	227.2003	227.2005	C_14_H_28_O_2_	1.9	0.72	1.02
n-Pentadecanoic acid	241.2144	241.2162	C_15_H_30_O_2_	3.9	0.72	1.02
Palmitic acid	255.2322	255.2318	C_16_H_32_O_2_	2.0	2.88	3.29
Hexyl 2-(4-hydroxy-3-methoxy-phenyl) acetate	265.1481	265.1434	C_15_H_22_O_4_	4.2	---	1.77
16-hydroxypalmitic acid	271.2257	271.2267	C_16_H_32_O_3_	2.6	6.12	1.52
Linoleic acid	279.2330	279.2318	C_18_H_32_O_2_	3.1		2.02
10,16-DHPA	287.2209	287.2216	C_16_H_32_O_4_	2.6	69.84	44.02
Heptadecanedioic acid	299.2228	299.2216	C_17_H_32_O_4_	3.4	2.88	1.26
8-hydroxyhexadecane dioic acid	301.2017	301.2009	C_16_H_30_O_5_	4.2	5.76	1.52
18-hydroxy-9S,10R-epoxy-octadecanoic acid	313.2387	313.2373	C_18_H_34_O_4_	3.9	---	9.36
9,10,18-trihydroxy-octadecanoic acid	331.2487	331.2479	C_18_H_36_O_5_	2.4	2.88	24.03
2,3-Divanillyl-1,4-butanediol	361.1563	361.1645	C_20_H_26_O_6_	4.5	2.16	4.05

[M − H]^−^_exact_: Molecular Weight exact, [M − H]^−^_obs_: Molecular Weight observed, % RA: % Relative Area. Error [ppm]: Absolute value of the deviation between measured mass and theoretical mass of the selected peak in [ppm].

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
