# Peer review of "Cutin from Solanum Myriacanthum Dunal and Solanum Aculeatissimum Jacq. as a Potential Raw Material for Biopolymers"

_polymers, 2020, doi:10.3390/polym12091945_

Round 1

Reviewer 1 Report

In this manuscript, the fruits of S. aculeatissimum and S. myriacanthum were analyzed, in order to find new renewable natural sources to design green and commercially available bioplastics. Some interesting results are presented. However, moderate revision is needed to further improve the manuscript. For instance,

The abstract and conclusion should be improved, and some values should be added.

In the introduction part, the authors should summarize the related attempts on the development of both natural and artificial sources.

The scientific principles should be further concluded in this work, and some pictures should be clearer.

There are some English language errors, which should be improved carefully.

Reviewer 2 Report

The proposal of the manuscript is important and it has merit for publication.

To improve your presentation, we suggest the authors:

In the introduction, the authors present the cutin as

“…cutin (a C6 and C18 long-chain hydroxy acid polyester),..”

Other authors present with C16-C18.

Lines 75-78:

“…the cuticle was treated with A. niger pectinase (EC 3.2.1.15) (10 mg/mL) for 1 week. After this, cell wall polysaccharides were removed with an enzymatic digestion using A. niger cellulose (EC 3.2.1.4) (80 mg/mL) for 1 week and A. niger hemicellulose (EC 3.2.1.4) (80 mg/mL) for 1 week…”

The enzyme activity was not detailed and the concentrations used for the solid substrates were expressed in mass / volume. Please express according to the substrate used.

Note the names of the enzymes: cellulase, hemi-cellulase?

Please rewrite the paragraph accordingly.

Lines 80-81:

“500 g of S. aculeatissimum peel yielded 386 g (77.2%) of cutin and 500 g of S. myriacanthum yielded 394 g (78.8%) of cutin.”

Please review these values: on a dry weight basis?

Do the peels have no other compounds: cellulose, hemicellulose and pectin?

Very high cutin yield.

Demonstrate how this yield was found.

Line 85: “TFA solution 2.0 M”

Please change to “TFA solution 2.0 mol L-1

Please correct the concentration units of “M” or “N” to mol L-1 throughout the manuscript.

Line 154: “figure 2.”

Authors must format the presentation of the figures: “figure”, “Fig.”, As required by the journal.

The authors do not explain the reader very well about the composition of the shells and the resulting cutin. Although studies on the isolates are well developed, there is a need to make it clear, the existing percentage and what is obtained after purification, from the peels of the fruits studied.

On the other hand, it would be appropriate to present a comparison of yield with cutin obtained, for example, from the peel of apples and citrus fruits.

Round 2

Reviewer 1 Report

Accept after minor revision.

Reviewer 2 Report

The authors followed the reviewers' suggestions. In the form presented, the manuscript can be indicated for publication.